# Role of the Green Husks of Persian Walnut (*Juglans regia* L.)—A Review

Laurine Kithi [1], Éva Lengyel-Kónya [2], Mária Berki [2] and Géza Bujdosó [1,*]

[1] Research Centre for Fruit Growing, Hungarian University of Agriculture and Life Sciences, 1223 Budapest, Hungary; laurineriziki28@gmail.com

[2] Institute of Food Science and Technology, Hungarian University of Agriculture and Life Sciences, 1118 Budapest, Hungary; lengyelne.konya.eva@uni-mate.hu (É.L.-K.); berki.maria@uni-mate.hu (M.B.)

* Correspondence: bujdoso.geza@uni-mate.hu

**Abstract:** Green husks are the outer layer of walnut fruits. They form part of the agro-residues discarded away upon nut maturity in the walnut industry. Although it is not used, research studies show that green husk is a rich source of natural bioactive phytochemicals. A total of 83 individual phenolic compounds were identified in walnut husks, mainly consisting of naphthoquinones, flavonols, and hydroxycinnamic acids. There is no standard profile of phenolic compounds in walnuts; the quantity and quality of phenolic compounds vary by cultivar. Walnut extracts exhibit strong antioxidant activities and play an important role in the plant's defence mechanisms against diseases, attacking different plant organs. The study provides a review of total phenolic content and individual phenolic compounds identified in green husks of different walnut cultivars as documented in different studies. It also explores the role and application of green husks in various industries such as traditional medicine, textile, wood, leather, beverage, and agriculture. In addition, the effects of phenolic compounds on biotic and abiotic factors are also evaluated.

**Keywords:** antioxidants; husk; extraction methods; nut phenology; phenolic compounds; walnut cultivars





## 1. Introduction

Persian or English walnut is one of the most important nut tree crops grown for its nutritious nuts, timber, and landscaping. It provides a source of employment and income to farming households and other value-chain players [1]. Walnut production is found in regions between latitudes 30–55° and 30–40° in the Northern and Southern Hemisphere, respectively. They include countries in Asia, Europe, North America, North Africa, South Africa, Australia, New Zealand, Chile, and Argentina [1].

In 2009, the total walnut production was estimated at 2.28 million tons of dried nuts with shells. A decade later, production had increased to 4.49 million tons. The top five producers include China, the United States of America, Iran, Turkey, and Mexico. China produced more than 50% of the total global production. The United States of America and Iran produced 592,390 tons and 321,074 tons of dried nuts with shells, respectively [1,2]. Annual production is steadily increasing in Central Europe [3]. In 2019, statistics show that Hungary's production was estimated at 6820 tons of dried nuts with shells [2].

Walnuts are mainly grown for their kernels and wood to a lesser extent; the other parts (hard shell, green husk) are produced as waste. The main products of walnuts are dried nuts with shells and kernels due to their nutritional value [4]. However, the walnut green husk is also a valuable part of the fruit due to its high phenolic compounds. In our review, we study the walnut green husk from a phytochemical point of view. Based on studies, there are different analytical methods for examining phenolic compounds in the green husks, and authors published data on a wide scale. Our aim is to document the various extraction methods that have been used before and the quantities of some phytochemicals established.

## 2. Botany of Walnut

Walnuts are large deciduous trees in the Juglandaceae family, arguably native to some countries in the Balkan Peninsula and Central Asia [5]. Mature trees are anchored on deep root systems, with big tap roots [6] and heavy, close-grained stems that produce valuable hardwood suitable for furniture. The trunk has large brown/copper glabrous branches with characteristically many-chambered pith. They also have broad canopies of about 18-m wide and 13-m high, and elongated pinnate compound leaves of 5–23 leaflets. Walnuts are commonly propagated vegetatively with grafting and budding, including in vitro culture [7]. Sexually propagated trees take 8–12 years to mature and produce nuts with many variations [8].

Flower differentiation and anthesis start, bearing monoecious flowers [9]. After meeting the chilling requirements that vary from 400 to 1500 h in temperatures between 0 °C and 7 °C [10,11]. This occurs from mid-April for early cultivars to mid-May for late-blooming cultivars [12]. Petal-less male drooping catkins develop on older branches, and clusters of 2–5 female flowers are borne on new terminal shoots or lateral spurs [13,14]. The flowers are heterodichogamous, some cultivars are protogynous, and others are protandrous. Imperfect overlap between pollen shedding and female flower receptivity necessitates the presence of two complementary cultivars to pollinate each other [1].

Walnut fruits consist of a kernel, seed coat, shell, and pericarp, commonly called green husk [15]. The semi-fleshy green pericarp covers a hard corrugated shell enclosing a four-celled edible nut. Walnut husks are composed of the epicarp, mesocarp, and endocarp supplying nutrients for the seed's growth [16]. The pericarp dehisces allow the seed to fall upon the maturity of nuts. The walnut seed contains two cotyledons enclosed in the shell with a thin lining called the pellicle. Selections of English walnuts are diverse, ranging from round to very elongated shapes, pea-sized to more than 50-mm shell diameter [8].

## 3. Nut Characteristics

### 3.1. Physical Characteristics of Kernel

Physical and phytochemical studies show significant kernel variations across walnut cultivars. Walnut fruits come in different dimensions, shapes, and kernel appearances, and have different content of chemical compounds. The main physical quality features are kernel weight and colour, shell shape and size, kernel ratio, and cracking ratio [17]. Others include sphericity, porosity, volume, bulk and true densities, coefficient of friction, and terminal velocity [18,19].

Table 1 shows the nut characteristics of cultivars bred in Europe. All cultivars reach 30-mm in diameter and at least 40% in kernel ratio (kernel weight/total nut weight). It is also interesting to see that there are no global cultivars in walnut-producing countries. Every country has its breeding program providing different assortments for local producers.

**Table 1.** Important locally bred European walnut cultivars.

| Country | Walnut Cultivars | Ripening/Harvest Period | Nut Size (mm) | Kernel Ratio (%) | Reference |
|---|---|---|---|---|---|
| Bulgaria | Dryanovski, Izvor 10, Petrushtinski, Sheinovo, Silistrenski | Second week September–first week October | 30–36 | 41–50 | [20,21] |
| Czech Republic | Apollo, Mars, Sychrov, Viktoria | Second week September–first week October | 32–34 | 42–52 | [22,23] |
| France | Feradam, Ferbel, Ferjean, Fernette, Fernor, Ferouette, Fertignac, Franquette, Lara | Third week September–second week October | 30–34 | 41–49 | [24] |
| Germany | Aufhausen Baden Walnussbaum, Alex, Börde Riesen Walnussbaum, Esterhazy II, Geisenheim Nr. 26, Geisenheim Nr. 120, Geisenheim Nr. 139, Kurmarker Walnuss Nr. 1247, Moselaner Walnussbaum Nr 120, Ockerwitzer lange, Red reif, Riesenpilar, Seifersdorfer runde, Spreewalder Walnuss Nr. 286, Weinsberg 1, Weinheimer Walnuss Nr. 139, Wunder von Monrepos | Third week September–second week October | 30–34 | 41–50 | [24,25] |
| Hungary | Alsószentivani 117, Alsószentiváni kései, Bonifác, Eszterházi kései, Érdió1, Milotai 10, Milotai bőtermő, Milotai intenzív, Milotai kései, Tiszacsécsi 83 | Second week September–first week October | 32–35 | 42–45 | [3,12,17] |
| Moldova | Kalarashsky, Kishinevsky, Kostjuzhensky, Korzheutsky, Skinossky | Third week September–second week October | 32–36 | 43–57 | [26] |
| Romania | Germisara, Jupâneşti, Redval, Sibişel 44, Timval, Unival, Valcor, Valcris, Valrex, Valmit, Verisval | First week September–first week October | 36–42 | 36–53 | [27–32] |
| Serbia | Backa, Champion, Kasni rodni, Milko, Macva, Rasna, Tica, Sampion, Sava, Srem | Second week September–first week October | 32–35 | 42–58 | [33–36] |
| Slovakia | Jupiter, Saturn | Second week September–first week October | 32–34 | 42–51 | [22] |
| Slovenia | Erjavec, Krka, Sava, Zdole-62 | Third week September–first week October | 29–32 | 41–54 | [37,38] |
| Turkey | Ahir Nut, Bilecik, Kaman 1, Kurtulus 100, Maras 18, Sebin, Sutyerez 1, Yalova 1, Yalova 3 | Second week September—first week October | 32–34 | 40–55 | [39–42] |
| Ukraine | Bukovynszky 1, Grozynetsky, Klishkivsky, Tsernivetsky 1 | Second–third week September | 32–35 | 41–55 | [43] |

*3.2. Chemical Compounds in Kernel*

Walnuts contain numerous phytochemical constituents such as polyphenols, fatty acids, mineral elements, vitamins, digestible proteins, amino acids, carbohydrates, and dietary fibre in their bark, roots, leaves, and fruits [44–49].

Polyphenolic compounds are secondary metabolites characterized by phenolic rings synthesized by plants as a defence against stress factors such as radiation, pest, and diseases. Classification of these compounds is based on their chemical structure. They include flavonoids, phenolic acids, stilbenes, lignans, and others [50,51]. A total of 83 phenolic compounds were identified in walnut husks, mainly consisting of naphthoquinones, flavonols, and hydroxycinnamic acids [52]. A related study by Medic et al. [53] identified and quantified 91 individual phenolic compounds. Forty-one of these were in root tissues, and others in petiole, bark, buds, and leaf tissue. They comprised 28 hydroxybenzoic acids, 22 naphthoquinones, 20 flavonols, 11 flavanols, 8 hydroxycinnamic acids, and 2 coumarins. Table 2 highlights polyphenolic compounds identified in walnut fruit in different studies.

**Table 2.** Identified compounds in walnuts.

| Polyphenol | Compound Name | Refs |
|---|---|---|
| Flavonoids | Quercetin, Myricetin, (+)-Catechin, (−)-Epicatechin, Rutin, Taxifolin, isoquercitrin, Kaempferol, 2,5-Dihydroxybenzoic acid, Naringenin | [3,52,54–56] |
| Phenolic acids | 4-Coumaric, Syringic, Caffeic, Ferulic, Protocatechuic, Sinapic, Quinic, Phtalic, Ellagic, Gallic, Vanillic, 2,5-Dihydroxybenzoic acid, Neochlorogenic, *p*-Hydroxybenzoic, *p*-Coumaric | [52,54,57–62] |
| Stilbenes | Resveratrol, pterostilbene, ε-viniferin, Trans-astringin | [51] |
| Lignans | Secoisolariciresinol, Pinoresinol, Lariciresinol, Matairesinol, Enterolactone. | [63–65] |
| Tannins | 2,3-hexahydroxydiphenoylglucose, Pedunculagin, 2,3,4,6-tetragalloylglucose, Casuarictin, Castalagin | [66,67] |
| Quinones | Juglone, Plumbagin, Lawsone, Emodin, Chrysophanol | [54] |

Kernels are rich in monounsaturated and polyunsaturated fatty acids such as alpha-linolenic acid (ALA), an omega-3 fatty acid, and linoleic acid (LA), an omega-6 fatty acid [68], oleic acid, and small amounts of saturated fatty acids. Studies show that its oil content ranges from 45.6% to 79.4% [5,15,18,45,49,69,70], with unsaturated fatty acids, tocopherols, and phytosterols being the most dominant. The main fatty acids present in walnuts include linoleic, oleic, linolenic, palmitic, and stearic acids [71–75]. Palmitoleic acid, arachidic acid, and eicosenoic acid are present in small amounts [72].

Kernels contain a significant amount of protein ranging from 10.6 to 25.0% [5,18,47,70,76]. Gu et al. [77] reported 40.0–45.0% protein content in *J. sigillata* kernels. Glutelin is the most abundant protein (72.1%); others (globulin, albumin, and prolamin) are present in smaller amounts [47]. These proteins come with various free amino acids, notably alanine, arginine, and glutamate [78].

Other biochemicals present in walnut kernels include carbohydrates (5.0–24.0%); minerals (phosphorus, potassium, and traces of sulphur); micro-elements (Ca, Mg, Fe, Cu, Zn, and I); vitamins: E, C, B1, B2, and A [5]; dietary fibre [49].

## 4. Walnut Green Husks

The green husk is a fleshy outer layer enclosing the shell of a nut [79]. Upon nut ripening, the husk gradually darkens. As a by-product, it is considered waste upon harvesting [13]. Sebahattin et al. [80] report that the average fresh husks-biomass to total walnut-biomass ratio is 57.15%. Husks are therefore produced in large amounts, and poor disposal lead to environmental pollution. However, they are a natural source of bioactive compounds, which can be used for diverse purposes.

## 4.1. Phenolic Compounds in Walnut Green Husks

Similar to other parts of the tree, numerous bioactive polyphenolic compounds associated with antioxidant and antimicrobial properties have been identified in green husks [81]. These compounds, as secondary metabolites, cover a large group of heterogeneous bioactive compounds. Their synthesis is regulated by different enzymes in different metabolic pathways, and they have several roles in fruit growth processes. The amount of these compounds depends on the genotype, environmental and climatic conditions as well as geographical conditions. Furthermore, it depends on the development stage of the fruit. It is necessary to follow up on these changes and consider the date of extraction when the amount reaches its maximum concentration [82]. The open green husks have higher antioxidant capacity and polyphenol content than the closed ones [83].

The basis of walnut husk utilization could be the high antioxidant activity of these phenolic compounds. The isolated and identified phenolic compounds are divided into different groups according to their chemical structure. Ellagic acid and tannic acid belong to hydrolyzable tannins, both present in walnut husks. Naphthoquinones, naphthoquinone glycosides, and naphthalenes are also isolated in green husks and widely studied. Juglone, the most important phenolic compound in the husk, is a naphthoquinone. Derivatives of juglone are also present. Other compound groups are $\alpha$-tetralones, $\alpha$-tetralones glycosides, and $\alpha$-tetralones dimers. The most significant hydroxybenzoic acids in husk are gallic acid, protocatechuic acid, syringic acid, vanillic acid, salicylic acid, 3,4-dihydroxybenzoic acid and 2,3-dihydroxybenzoic acid, and benzoic acid. Caffeic acid, ferulic acid, chlorogenic acid, p-coumaric acid, sinapic acid, chlorogenic acid, and trans-ferulic acid are also characterized in green husks, which make the group of hydroxycinnamic acids. Flavonoids are represented in the husk by (+)-catechin, (−)-epicatechin, myricetin, quercetin, sudachitin, cirsilineol, and 5,6,4′-trihydroxy-7,3′-dimethoxy-flavone, apigenin, eriodictyol, kaempferol, rutin. Other secondary metabolites, such as diarylheptanoids, are isolated and identified as well, e.g., rhoiptelol, juglanin A, juglanin B, and juglanin C [15].

## 4.2. Extraction Methods of Phenolic Compounds

The choice of extraction solvent is a critical consideration because of its chemical variability and complexity. Many research groups have investigated the effects of different extraction solvents and how the development stage affects the content of phenolics as well. A negative correlation was established between growth and the concentration of the phenolic compounds [82]. According to the investigation by Zhang [84], extraction by methanol, ethanol, and acetone shows the highest content of phenolic compounds and antioxidant activity compared to ethyl-acetate and water. Hexane yielded the least number of phenolic compounds. Fernandez-Agulló et al. [81] investigated the extraction yield of methanol, ethanol, methanol/water 50/50, and ethanol/water 50/50. It was established that ethanol/water 50/50 resulted in higher bioactivity and that the antioxidant properties depend on the concentration of the extraction solvent. In a related study, Hama et al. [85] used methanol (80%) for phenolic extraction with chloroform, ethyl acetate, and n-butanol. Ethyl acetate yielded the highest amount of phenolics, followed by chloroform and n-butanol, respectively.

A comparative study of antioxidant activity and individual phenolic compounds in green husk extracts using three different solvents (70% ethanol, 40% ethanol, and 40% ethanol/sugar 50/50) was performed by Cosmolescu and co-workers [86]. In the case of 70% ethanol, it was found that the concentrations of gallic, vanillic, chlorogenic, caffeic, syringic, salicylic, ellagic acids, juglone, catechin, epicatechin, myricetin, and quercetin were the highest. In the case of 40% ethanol, ferulic acid and rutin were more, compared to the other compounds mentioned above. Forty per cent ethanol and sugar (the traditional way of walnut liqueur production) resulted in the highest amount of rutin. Jakopic and co-workers [87] examined the effect of different ethanol concentrations on total phenolic concentration and the number of individual compounds. Generally, they found that increasing ethanol concentration resulted in increased phenolic concentrations. However,

in some cases (gallic, chlorogenic, vanillic, and syringic acid, (+)-catechin, and juglone), they achieved better extraction yield with 40% ethanol. In another study, methanol gave better extraction for juglone, (+)-catechin, gallic, protocatechuic, and chlorogenic acids, whereas ethanol resulted in higher amounts of ellagic and sinapic acids [88].

Barekat et al. [89] established that phenolic compounds in walnut husks could have significant antioxidant power ranging from 256.5 to 746.8 score g$^{-1}$ dry weight (DW) by using the PAOT (total antioxidant power) method. This method uses two specific electrodes to measure the changes in the electrochemical potential in the reaction medium. These changes are in correlation with the antioxidant properties, and results are expressed in PAOT score (total antioxidant power) per gram of dw walnut husk. The identified phenolic compounds include tannins, flavonoids, stilbenes, lignans, quinones, diarylheptanoids, and phenolic acids [44,51,65,90]. Sheng et al. [44] reported specific phenolic compounds identified in walnut husks. Tables 3 and 4 below detail more studies, the extraction methods used, applied analytical methods, and the content of the phenolic compound established thereof.

**Table 3.** Summary of studies and overall antioxidant properties in walnut green husks.

| Plant Materials, Place of Study | Extraction Solvent | Methods | Results | Reference |
|---|---|---|---|---|
| Green husk, Iran | 80% ethanol in bi-distilled water | Total phenolic content by Folin–Ciocalteu spectrophotometric method | 35.2–59.8 mg GAE/g DW | [89] |
| Green husk, Hungary | 80% methanol in bi-distilled water | Total phenolic content by Folin–Ciocalteu spectrophotometric method | 44.2–57.4 mg GAE/g DW | [3] |
| Kernels, leaves, husk, and bark—Morocco | Methanol | Total phenolic content by Folin–Ciocalteu spectrophotometric method | 306.36 ± 4.74 mg GAE/g dried husk extract | [91] |
| Green husk, China | 50% methanol in bi-distilled water | Total phenolic content by Folin–Ciocalteu spectrophotometric method | 0.54–1.33 mg GAE/g FW | [44] |
| Green husk, Chile | Methanol, ethanol | Total phenolic content by Folin–Ciocalteu spectrophotometric method | 1862.9 ± 72.4 mg GAE/100 g DW extracted lyophilized green husk by ethanol | [92] |
| Green husk, Portugal | Water | Total phenolic content by Folin–Ciocalteu spectrophotometric method | 40.39–84.46 mg GAE/g extract | [81] |
| Green husk, Iran | Methanol | Total phenolic content by Folin–Ciocalteu spectrophotometric method | 99.98–122.26 mg GAE/g extract | [93] |
| Green husk, Iran | Acetone/water (*v/v*, 70/30) | Total phenolic content by Folin–Ciocalteu spectrophotometric method | 95.2 ± 6.29 mg GAE/g DW | [94] |
| Green husk, Iran | Methanol | Total phenolic content by Folin–Ciocalteu spectrophotometric method | 15.15–108.11 mg GAE/g extract | [95] |
| Green husk, Iran | Methanol | Total phenolic content by Folin–Ciocalteu spectrophotometric method | 19.61–36.10 mg GAE/g DW | [96] |
| Green husk, Iran | 80% ethanol in bi-distilled water | Antioxidant Activity by DPPH Assay | IC50: 146.8–249.3 µg/mL | [89] |
| Kernels, leaves, husk, and bark—Morocco | Methanol | Antioxidant Activity by DPPH Assay | IC50: 32.27 ± 0.69 µg/mL husk extract | [91] |
| Green husk, Chile | Methanol, ethanol | Antioxidant Activity by DPPH Assay | 2663.2 ± 154.6 mg Trolox Equivalent/100 g DW extracted lyophilized green husk by ethanol | [92] |
| Green husk, Portugal | Methanol, ethanol and 50% aqueous solutions of methanol and ethanol | Antioxidant Activity by DPPH Assay | EC50: 0.33–0.72 mg/mL | [81] |

**Table 3.** *Cont.*

| Plant Materials, Place of Study | Extraction Solvent | Methods | Results | Reference |
|---|---|---|---|---|
| Green husk, Iran | Methanol | Antioxidant Activity by DPPH Assay | 76.71–89.81% | [93] |
| Green husk, Iran | Acetone/water (*v*/*v*, 70/30) | Antioxidant Activity by DPPH Assay | 85 $\pm$ 1.6 µg/mL | [95] |
| Green husk, Iran | Methanol | Antioxidant Activity by DPPH Assay | IC50: 122–302 µg/mL | [95] |
| Green husk, Iran | Methanol | Antioxidant Activity by DPPH Assay | EC50: 0.25–0.40 mg/mL | [96] |
| Kernels, leaves, husk, and bark—Morocco | Methanol | Total flavonoid content by colorimetric method | 66.07 $\pm$ 2.68 mg RE/g dried husk extract | [91] |
| Green husk, China | 50% methanol in bi-distilled water | Total flavonoid content by colorimetric method | 0.34–1.01 mg RE/g FW | [44] |
| Green husk, Iran | Methanol | Total flavonoid content by colorimetric method | 16.71–49.00 mg CE/g | [94] |
| Green husk, Iran | Acetone/water (*v*/*v*, 70/30) | Total flavonoid content by colorimetric method | 65.2 $\pm$ 5.53 mg CE/g DW | [95] |
| Green husk, Iran | Methanol | Total flavonoid content by colorimetric method) | 3.59–22.91 mg QE/g extract | [97] |
| Green husk, Iran | Methanol | Total flavonoid content by colorimetric method) | 534–1064 mg CE/100 g DW | [96] |
| Kernels, leaves, husk, and bark—Morocco | Methanol | Antioxidant activity ABTS method | IC50: 145.86 $\pm$ 1.61 µg/mL husk extract | [91] |
| Kernels, leaves, husk, and bark—Morocco | Methanol | Ferric ion-reducing power was determined by the FRAP method | IC50: 10.45 $\pm$ 0.59 µg/mL husk extract | [91] |
| Green husk, Iran | Reaction media containing a free radical molecule called intermediate (M•) | Total Antioxidant Power by PAOT Technology | 256.52–746.88 PAOT score/g dw | [89] |
| Green husk, Chile | Methanol, ethanol | Antioxidant capacity by ORAC method | 44,920 ORAC units–µmol Trolox Equivalent/100 g DW extracted lyophilized green husk by ethanol | [92] |

UHPLC-PDA-HRMS/MS—ultra-high-performance liquid chromatography with photodiode array and high-resolution mass spectrometer, HPLC-ESI-DAD—high-performance liquid chromatography with diode array detection and with electrospray ionization mass spectrometry, HPLC UV-Vis—high-performance liquid chromatography with ultraviolet-visible region detection, PAOT—total antioxidant power, ABTS—2,2′-azino-bis(3-ethylbenzothiazoline-6-sulfonic acid), DPPH—2,2-diphenyl-1-picrylhydrazyl, FRAP—ferric reducing antioxidant power, ORAC—oxygen radical absorbance capacity, DW—dry weight, FW—fresh weight, IC50—half-maximal inhibitory concentration, EC50: the concentration required to provide 0.5 of absorbance, GAE—gallic acid equivalent, CE—catechin equivalent, QE—quercetin equivalent, RE—rutin equivalent.

**Table 4.** Summary of chromatographic methods and phenolic compounds detected in walnut green husks.

| Plant Materials, Place of the Study | Extraction Solvent | Chromatographic Analysis | Results | Reference |
|---|---|---|---|---|
| Green husk, Iran | 80% ethanol in bi-distilled water | UHPLC-PDA-HRMS/MS, Hichrom Alltima C18 column (150 × 2.1 mm − 5 μm), mobile formic acid (0.1%) in water for A and formic acid (0.1%) in acetonitrile for B, linear gradient, then isocratic | Minoxidil, myricetin, quercetin 4′-glucoside, taxifolin, quercetin pentoside, catechin, abscisic acid, salicylate glucuronide, neochlorogenic acid, taxifolin 7-glucoside, gallic acid derivative | [89] |
| Green husk, Hungary | Bi-distilled water/2% acetic acid in methanol, 30/70 | HPLC-ESI-DAD, column: Sphinx 5 μm 250 × 4.6 mm, mobile phase: (A) 0.1% formic acid in bi-distilled water and (B) 0.1% formic acid in acetonitrile in gradient elution mode | Hydroxybenzoic acids, hydroxycinnamic acids, flavonoids, naphthoquinones, juglone | [3] |
| Kernels, leaves, husk, and bark—Morocco | Methanol | HPLC-ESI-DAD-MS/MS, The column was a Poroshell 120 EC-C1, C18 (150 × 2.1) mm × 5 μm. The mobile phase was (A) 0.1% formic acid in the water, (B) acetonitrile. The established elution gradient was isocratic | Dihydroxybenzoic acid derivative, acacetin aglycone, caffeoyl-D-glucose, quercetin O hexoside 1, apigenin-7-O-glucoside, caffeoyl derivative, p-coumaroyl derivative, caffeoyl hexose-deoxyhexoside, quercetin pentoside | [91] |
| Green husk, China | 50% methanol in bi-distilled water | HPLC, Agilent SB-C18 column (250 × 4.6 mm, 5 μm) the mobile phase consisted of water containing 0.5% acetic acid as eluent A and methanol as eluent B. gradient elution | Gallic acid, neochlorogenic acid, (+)-catechin, p-hydroxybenzoic acid, chlorogenic acid, vanillic acid, caffeic acid, epicatechin, syringic acid, p-coumaric acid, ferulic acid, o-coumaric acid, rutin, myricetin, quercetin and juglone | [44] |
| Green husk, Chile | Methanol, ethanol | Juglone by HPLC-PDA, column: Kinetex Evo® C18 100 A 5 μm (250 mm × 4.6 mm). A gradient of solvents was used as the mobile phase: solvent A was 2.0% acetic acid in aqueous solution and solvent B was 0.5% acetic acid in aqueous solution and acetonitrile (1:1 ratio) | 169.1 mg/100 g DW | [92] |
| Green husk, Portugal | 80% ethanol in bi-distilled water | HPLC-DAD-ESI/MS | Naphthalene derivatives (including tetralone derivatives), phenolic compounds (hydroxycinnamic acids and flavonols) | [97] |

**Table 4.** *Cont.*

| Plant Materials, Place of the Study | Extraction Solvent | Chromatographic Analysis | Results | Reference |
|---|---|---|---|---|
| Green husk, Iran | Methanol | HPLC UV-Vis, column: Eurospher 100-5 C18 column (25 cm × 4.6 mm; 5 μm), mobile phase: purified water with 2% acetic acid (A) and acetonitrile (B), isocratic and linear gradient mode | Ascorbic acid, gallic acid, rutin, caffeic acid, p-hydroxy benzoic acid, vanillic acid, p-cumaric acid, syringic acid, ferulic acid, sinapic acid | [94] |
| Green husk, Iran | Methanol | HPLC UV-Vis, column: Eurospher 100-5 C18 column (25 cm × 4.6 mm; 5 μm), mobile phase: purified water with 2% acetic acid (A) and acetonitrile (B), isocratic and linear gradient mode | Vanillic acid, 1-naphthol, caffeic acid, salicylic acid, 8-hydroxyquinoline, tannic acid | [96] |

UHPLC-PDA-HRMS/MS—ultra-high-performance liquid chromatography with photo diode array and high-resolution mass spectrometer, HPLC-ESI-DAD—high-performance liquid chromatography with diode array detection and with electrospray ionization mass spectrometry, HPLC UV-Vis—high-performance liquid chromatography with ultraviolet-visible region detection, PAOT—total antioxidant power, ABTS—2,2′-azino-bis(3-ethylbenzothiazoline-6-sulfonic acid), DPPH—2,2-diphenyl-1-picrylhydrazyl, FRAP—ferric reducing antioxidant power, ORAC—oxygen radical absorbance capacity, DW—dry weight, FW—fresh weight, IC50—half-maximal inhibitory concentration, EC50: the concentration required to provide 0.5 of absorbance, GAE—gallic acid equivalent, CE—catechin equivalent, QE—quercetin equivalent.

As summarised in Tables 3 and 4, not only the overall antioxidative properties are in the focus of research studies, but also the investigation of individual phenolic compounds behind these bioactive characteristics. Results of the total phenolic content by the Folin–Ciocalteu method, the antioxidant activity by DPPH assay, the total flavonoid content by colorimetric method and other methods (ABTS, FRAP, ORAC, PAOT) show that walnut green husks have considerable antioxidant characteristics. The compounds identified by chromatographic measurements show the individual molecules behind the antioxidative effects as well. As indicated in Tables 3 and 4, different extraction solvents provide maximum yield at different concentrations. The studies show that total phenolic content can either be as high as 306.36 mgGAEg$^{-1}$DW [91] or as low as 3.72 mgGAEg$^{-1}$DW [93]. The study by Sheng et al. [44] revealed a decline in phenolic content from nut-bearing to the maturity stage. Conversely, flavonoid content increased. A chemical analysis of seven genotypes ('KZ7', 'KZ9', 'KZ15', 'OR126', 'Sebin', 'Pedro', and 'Chandler') by Rahmani et al. [94] exhibited different antioxidant potentials attributed to their differences in phenolic content. The total phenolic content ranged from 99.98 to 122.26 mgGAEg$^{-1}$, whereas the flavonoid content ranged from 16.71 to 49.0 mgCEg$^{-1}$. Similar variations were also observed on Hungarian-bred cultivars. Bujdosó et al. [3] established that 'Bonifác' had the highest phenolic content; however, their nuts were very susceptible to *Xanthomonas arboricola* pv. *juglandis*.

Ghasemi et al. [95] observed variances in phenolic and flavonoid content in husks with changes in altitude and temperature. A positive correlation between total phenolic and flavonoid content with altitude was observed; however, the correlation with the temperature was negative [95]. Ghesami's team attributed the significant differences in phenolic and flavonoid content to the differences in geographical and climatic conditions where samples were obtained.

Mikulic-Petkovsek et al. [98] observed a decline in total phenolic content concerning the physiological stage of the nuts. However, it was noted that the biosynthesis of phenolic compounds in healthy plants may be triggered by pathological infections. *Xanthomonas arboricola* pv. *juglandis* infected tissues of the husk exhibited higher content of hydroxycinnamic acids, gallic acid, quercetin, and catechin than uninfected tissues. Similar observations were made by Sheng et al. [44]. The total phenolic and flavonoid content in husks declined with increased maturity to subsequent ripening of the walnut fruit. The differences in phenolic content reported by the research groups [44,98] may be related to the sample origin, the genotype, the ripening status of the samples, the applied solvent, and measurement methods.

### 4.3. Uses of Walnut Husks

Despite being underutilized, green husks remain an important agricultural by-product and a natural source of bioactive compounds. Like other parts of the walnut tree, the green husk has been used in traditional medicine to relieve pain and treatment of cancer, diabetes, microbial infections and skin and heart diseases [15,79]. The walnut green husk is a source of glucans and pectins, which are useful in human pathology treatment and induce defence mechanisms to respond to wounding [15]. The husks contain juglone, which is a natural dye used in the textile and wood industry, as well as tanning leather [83,99]. Allelopathic compounds in husks serve as pesticides and herbicides [100,101]. It is also important to note that green husks are used in making wine, traditional liqueur, and jam [102,103], whereas green nuts are used in pharmaceuticals and cosmetics [81].

### 4.4. Effects of Walnut Green Husk Biochemical Compounds on Abiotic and Biotic Factors

The numerous bioactive compounds present in walnut husks are associated with both positive and negative effects on abiotic and biotic factors such as soil, water, microorganisms, and insects, including herbivores. They play a significant role in human, animal, and plant health because these compounds have antioxidant and antiradical properties. They exhibit

anticarcinogenic, antimutagenic, anti-inflammatory, cardioprotective, astringent, antiseptic, anthelmintic, and antiaging effects that are beneficial to human and animal health [45,104–108].

According to the literature, these compounds have antimicrobial and antifungal properties that prevent and control plant diseases [109]. They inhibit the initiation and progress of *X. arboricola* pv. *juglandis*, the walnut blight disease-causing bacteria [3] and pathogenic fungi such as *Botrytis cinerea*, *Alternaria alternata*, *Fusarium culmorum*, *Rhizoctonia solani*, and *Phytophthora cactorum* [110]. Both 4-coumaric acid and *p*-hydroxybenzoic acid, at minimum concentrations of 1 $gL^{-1}$ and 2 $gL^{-1}$, respectively, inhibit the growth of *Colletotrichum gloeosporioides* fungi responsible for walnut anthracnose disease [109]. Maleita et al. [111] established that at 50 ppm, 1,4-Naphthoquinone can cause 42% mortality in the juvenile root-knot nematode (*Meloidogyne hispanica*), thus a potential alternative to synthetic nematicides. Walnut and tea intercrops improved the availability of soil micronutrients such as potassium, nitrogen, phosphorus, and organic matter, including enhanced microbial diversity [112].

Besides the beneficial effects, the compounds have allelopathic and toxic effects on some plants and marine life [102,113,114]. Medic and his research team [113] evaluated the effect of juglone and other walnut allelochemicals and established allelopathic effects on snack cucumber at 1 mM concentration. Kocacaliskan & Terzi [115] reported strong seedling germination and seedling growth inhibition effect in tomatoes, cucumber, garden cress, and alfalfa. In the review by Liu et al. [102], juglone and plumbagin compounds present in walnut husks were highly toxic to some fish species. Hence walnut husks can be a threat to the environment if not well disposed of.

## 5. Conclusions

Walnut is an important tree crop not only for its valuable primary products but also its by-products, such as the husks. Although its potential has not been fully explored, walnut husks contain enormous natural bioactive compounds that are useful in various industries. Natural biochemicals present a safe alternative to synthetic antioxidants in modern medicine and food industries. Considering the significant quantities of husks, the biochemical compounds depend on different factors such as the cultivars, sample collection period inside the vegetation season, location of the orchard (latitude), and the method of examination. The phenolic compound content varies from 3.72 to 122.26 mg GAE $g^{-1}$, and the flavonoid concentration was in the range of between 0.34 and 66.07 $REg^{-1}$ based on the collected studies. The methods of examining bioactive compounds should be standardized to be able to compare the results.

**Author Contributions:** Conceptualization, L.K., É.L.-K., M.B. and G.B.; writing—original draft preparation, L.K. and M.B.; writing—review and editing, L.K., É.L.-K., M.B. and G.B.; supervision, É.L.-K. and G.B. All authors have read and agreed to the published version of the manuscript.

**Funding:** The first author received the Stipendium Hungaricum Scholarship (it ID: 2022-573581), which supported her to prepare this paper.

**Data Availability Statement:** All data were collected from the published research papers.

**Acknowledgments:** Thank you for the support to Stipendium Hungaricum Scholarship (grant number 2022-573581).

**Conflicts of Interest:** The authors declare no conflict of interest.

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
