# Peer review of "Role of the Green Husks of Persian Walnut (Juglans regia L.)—A Review"

_horticulturae, doi:10.3390/horticulturae9070782_

Round 1

Reviewer 1 Report

The submitted manuscript: "Role of the green husks of Persian walnut (Juglans regia L.) – a review" represents a brief review for the phytochemical composition and biological activities of walnut husks, which are generally considered as a by-product in the industry.  

The following changes are recommended and some clarifications should be made:

- The abstract section should be revised, since it is too generally written. The authors did not provide some key points in the abstract, such as the brief representation of walnut characteristics, phytochemical profile of walnut husks or their biological activities.

- The keywords should be revised. The terms “husk” and “walnut or Juglans regia L.” are missing.

- Table 2, please, check the name Gentsic acid.

- The authors should mentioned the antioxidant assay that was used in this citation, as well the expression results with the used equivalents: “Barekat et al. [87] established that phenolic compounds in walnut husks could have significant antioxidant power ranging from 256.5 to 746.8 score g-1 dry weight (DW).”

- Table 3 should be revised, particularly in the column for Material and methods. The chromatographic analysis should be presented separately from the Folin Ciocalteu method for total phenolics or aluminium chloride method for total flavonoids. Why the authors presented the reaction mixture for total phenolics, while chromatographic analyses are only mentioned? The chromatographic analyses should include for example: type of column, mobile phase used etc. It is not clear if the extraction is used for total phenolics or for chromatographic analyses. I suggest clear representation of this column for the Material and methods with the following topics: plant material, extraction, Type of analysis (spectrophotometric for total phenolics and flavonoids or chromatographic analysis) with key components of the methods.

- All the abbreviations in Table 3 should be defined at the end of table as a note. (For example, GAE, RE, HPLC-DAD-ESI-MS, PDA-HRMS etc).

- Please check the following in the Table 3: Calorimetry?

- Please rephrase the following sentences:

1. “The study by Sheng et al. reveals that phenolic content in husks reduces from fruit-bearing to the maturity stage conversely, flavonoid content increases.”

2. A positive correlation between total phenolic and flavonoid content with altitude and a negative correlation with temperature [95].

3. In their review, [100]. indicate that juglone and plumbagin, compounds present in walnut husks are highly toxic to nine fish species.

- The section Conclusions should be improved.

Extensive editing of the English language is needed.

Author Response

Dear Reviewer 1,

first of all the authors take the opportunity to thank you for your time and your valueable comments. All of your comments helped us a lot to improve the quality of the manuscript.

Here you can read our answers to your questions:

Reviewer 1: The abstract section should be revised, since it is too generally written. The authors did not provide some key points in the abstract, such as the brief representation of walnut characteristics, phytochemical profile of walnut husks or their biological activities.

Answer: The abstract was revised, all of your requirements were involved.

Reviewer 1: The keywords should be revised. The terms “husk” and “walnut or Juglans regia L.” are missing.

Answer: The keyword were revised, both of your suggestions were incorporated.

Reviewer1: Table 2, please, check the name Gentsic acid.

Answer: It was checked and revised.

Reviewer1: The authors should mentioned the antioxidant assay that was used in this citation, as well the expression results with the used equivalents: “Barekat et al. [87] established that phenolic compounds in walnut husks could have significant antioxidant power ranging from 256.5 to 746.8 score g-1 dry weight (DW).”

Answer: Thank you for the comment. The mentioned sentence was revised.

Reviewer 1: Table 3 should be revised, particularly in the column for Material and methods. The chromatographic analysis should be presented separately from the Folin Ciocalteu method for total phenolics or aluminium chloride method for total flavonoids. Why the authors presented the reaction mixture for total phenolics, while chromatographic analyses are only mentioned? The chromatographic analyses should include for example: type of column, mobile phase used etc. It is not clear if the extraction is used for total phenolics or for chromatographic analyses. I suggest clear representation of this column for the Material and methods with the following topics: plant material, extraction, Type of analysis (spectrophotometric for total phenolics and flavonoids or chromatographic analysis) with key components of the methods.

Answer: Table 3 was revised.

Reviewer 1: All the abbreviations in Table 3 should be defined at the end of table as a note. (For example, GAE, RE, HPLC-DAD-ESI-MS, PDA-HRMS etc).

Answer: All abbreviations were checked and revised.

Reviewer 1: Please check the following in the Table 3: Calorimetry?

Answer: This word was revised.

Reviewer 1: Please rephrase the following sentences:

  1. “The study by Sheng et al. reveals that phenolic content in husks reduces from fruit-bearing to the maturity stage conversely, flavonoid content increases.”
  2. A positive correlation between total phenolic and flavonoid content with altitude and a negative correlation with temperature [95].
  3. In their review, [100]. indicate that juglone and plumbagin, compounds present in walnut husks are highly toxic to nine fish species.

- The section Conclusions should be improved.

Answer: The mentioned sentences were revised.

Comments on the Quality of English Language

Reviewer1: Extensive editing of the English language is needed.

Answer: The grammatical errors were corrected. .

Once again, the authors thank you for your time, suggestions, and efforts to improve the quality of the paper.

Yours sincerely,

the authors

Reviewer 2 Report

Manuscript title: Role of the green husks of Persian walnut (Juglans regia L.) – a review

This study has collective info on walnut botany plan and walnut green husk …………... However, revisions are necessary for the current version of the manuscript. The following questions to be addressed/considered may be helpful to improve the manuscript.

Major comments

·         Please make sure you have a line number throughout the manuscript.

·         In the title is missing, what kind of review this paper is? Critical review? Meta-analysis review? Or just viewpoint

·         Insufficient Abstract: In the abstract, the main aim and background of the manuscript are missing, the current version it only highlights the result. In addition, it would be even better to have a sentence as a future perspective.

·         Line 37-38, the aim or hypothesis of the study is not clear, nor the approach ….consider elaborating on what is this review about and why it is important?

·         Lake of scientific literature to support the statements and findings throughout the manuscript…... I have made some suggestions for that and more need it….

·         All TABLES need to be simplified, the current version of the tables is overcrowded please consider simplifying the tables so the readers can follow them easily…..

·         ·         I have a major concern about sections 2-4. The discussions are very descriptive!  The authors describe the results and compare the results a few with previous studies, for example, for green husk uses, the detail regarding the application is missing………..

Specific comments:

A reference is needed here, for example, you can use:

Introduction:

Line 27-30: A reference is needed here, for example, you can use:

https://doi.org/10.1007/978-981-15-7470-2_20

Line 75-78: I am not sure what you mean here. What about Californian or Chinese cultivars? These cultivars have been already recorded and validated – please check.

Line 27-33: A complicated sentence, please revise and check the grammar

Page 2 and 6: These are rather long sentences, better to break them down into more sentences.

Page 7, paragraph 5: A reference is needed here, for example, you can use:

https://doi.org/10.1080/22297928.2016.1152912

These sections are repeating information already presented and explain things in an unnecessarily complicated way. The quality of the manuscript would benefit from the whole section being condensed, Page 8; paragraphs 1 and 2, page 16; paragraph 1 and 3

Conclusion

The section should not be a summary of your study or an extension of the discussion. This section should illustrate the mechanistic links of findings of this study. The conclusions should answer the hypothesis of your study and should focus on the implication of your findings. Remember that the conclusions must be self-explanatory. This section should still highlight the novelty and implication of your study also.

Grammar and punctuation issuers need to be addressed. I have selected/mentioned some as examples.

Author Response

Dear Reviewer 2,

first of all the authors take the opportunity to thank you for your time and your comments. All of your comments helped us a lot to improve the quality of the manuscript.

Here you can read our answers to your questions:

Reviewer 2: Please make sure you have a line number throughout the manuscript.

Answer: The line numbering is not appearing in some MS versions. Based on our experience under Win10 it cannot be seen.

Reviewer 2: In the title is missing, what kind of review this paper is? Critical review? Meta-analysis review? Or just viewpoint

Answer: Based on the MDPI template the authors are not obliged to select the type of the review. For it is a viewpoint review.

Reviewer 2: Insufficient Abstract: In the abstract, the main aim and background of the manuscript are missing, the current version it only highlights the result. In addition, it would be even better to have a sentence as a future perspective.

Answer: The abstract was revised.

Reviewer 2: Line 37-38, the aim or hypothesis of the study is not clear, nor the approach ….consider elaborating on what is this review about and why it is important?

Answer: The aim was revised.

Reviewer 2: Lake of scientific literature to support the statements and findings throughout the manuscript…... I have made some suggestions for that and more need it….

Answer: Thank you for the additional references, both were useful and were happy to add them.

Reviewer 2: All TABLES need to be simplified, the current version of the tables is overcrowded please consider simplifying the tables so the readers can follow them easily…..

Answer: All Tables were revised.

Reviewer 2: I have a major concern about sections 2-4. The discussions are very descriptive!  The authors describe the results and compare the results a few with previous studies, for example, for green husk uses, the detail regarding the application is missing………..

Answer: Currently, the green husk is an agricultural waste, therefore there are just very few special uses.

Specific comments:

Reviewer 2: A reference is needed here, for example, you can use:

Introduction:

Line 27-30: A reference is needed here, for example, you can use:

https://doi.org/10.1007/978-981-15-7470-2_20

Answer: Thank you for sending us the reference, it was added to the main text.

Reviewer 2: Line 75-78: I am not sure what you mean here. What about Californian or Chinese cultivars? These cultivars have been already recorded and validated – please check.

Answer: We were focusing on the European bred cultivars.

Reviewer 2: Line 27-33: A complicated sentence, please revise and check the grammar

Answer: The sentence was revised.

Reviewer 2: Page 2 and 6: These are rather long sentences, better to break them down into more sentences.

Answer: The sentences were revised.

Reviewer 2: Page 7, paragraph 5: A reference is needed here, for example, you can use:

https://doi.org/10.1080/22297928.2016.1152912

Answer: Thank you for sending us the reference, it was added to the main text.

Reviewer 2: These sections are repeating information already presented and explain things in an unnecessarily complicated way. The quality of the manuscript would benefit from the whole section being condensed, Page 8; paragraphs 1 and 2, page 16; paragraph 1 and 3

Answer: The authors revised the mentioned parts.

Conclusion

Reviewer 2: The section should not be a summary of your study or an extension of the discussion. This section should illustrate the mechanistic links of findings of this study. The conclusions should answer the hypothesis of your study and should focus on the implication of your findings. Remember that the conclusions must be self-explanatory. This section should still highlight the novelty and implication of your study also.

Answer: The conclusion was revised.

Comments on the Quality of English Language

Reviewer 2: Grammar and punctuation issuers need to be addressed. I have selected/mentioned some as examples.

Answer: Grammar was corrected.

Once again, the authors thank you for your time, suggestions, and efforts to improve the quality of the paper.

Yours sincerely,

the authors

Round 2

Reviewer 1 Report

The authors satisfactorily responded to all my queries. 

Minor editing of English language is necessary.

Reviewer 2 Report

The revised manuscript has improved compared to the original version. The authors tried to address my questions as much as possible. I recommend the manuscript to be published!

Overall, the English language and grammar quality are good and acceptable.